# WFCRL: A Multi-Agent Reinforcement Learning Benchmark for Wind Farm Control

**Claire Bizon Monroc**
Inria and DI ENS, École Normale Supérieure, PSL Research University, Paris, France
IFP Energies nouvelles
`claire.bizon-monroc@inria.fr`

**Ana Bušić**
Inria and DI ENS, École Normale Supérieure, PSL Research University
Paris, France

**Donatien Dubuc**
IFP Energies nouvelles
Solaize, France

**Jiamin Zhu**
IFP Energies nouvelles
Rueil-Malmaison, France

## Abstract

The wind farm control problem is challenging, since conventional model-based control strategies require tractable models of complex aerodynamical interactions between the turbines and suffer from the curse of dimension when the number of turbines increases. Recently, model-free and multi-agent reinforcement learning approaches have been used to address this challenge. In this article, we introduce WFCRL (Wind Farm Control with Reinforcement Learning), the first open suite of multi-agent reinforcement learning environments for the wind farm control problem. WFCRL frames a cooperative Multi-Agent Reinforcement Learning (MARL) problem: each turbine is an agent and can learn to adjust its yaw, pitch or torque to maximize the common objective (e.g. the total power production of the farm). WFCRL also offers turbine load observations that will allow to optimize the farm performance while limiting turbine structural damages. Interfaces with two state-of-the-art farm simulators are implemented in WFCRL: a static simulator (FLORIS) and a dynamic simulator (FAST.Farm). For each simulator, 10 wind layouts are provided, including 5 real wind farms. Two state-of-the-art online MARL algorithms are implemented to illustrate the scaling challenges. As learning online on FAST.Farm is highly time-consuming, WFCRL offers the possibility of designing transfer learning strategies from FLORIS to FAST.Farm.

## 1 Introduction

The development of wind energy plays a crucial part in the global transition away from fossil energies, and it is driven by the deployment of very large offshore wind farms [44, 32]. Significant gains in wind energy production can be made by increasing the amount of wind power captured by the farms [32]. The power production of a wind farm is greatly influenced by wake effects: an operating upstream turbine causes a decrease in wind velocity and an increase in wind turbulence behind its rotor, which creates sub-optimal wind conditions for other wind turbines downstream. An illustration of this phenomenon can be seen on Figure 1. Wake effects are a major cause of power loss in wind farms, with the decrease in power output estimated to be between $10\%$ and $20\%$ in large offshore

38th Conference on Neural Information Processing Systems (NeurIPS 2024) Track on Datasets and Benchmarks.

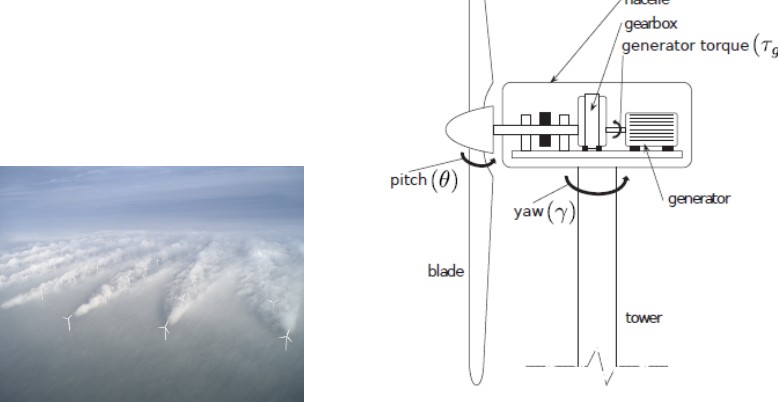

Figure 1: Left: Wake effects in the offshore wind farm of Horns Rev 1 - Vattenfall. Right: Schema of a wind turbine [6]. The *pitch*, *yaw* or *torque* can be controlled.

wind farms [4]. Higher turbulence in wakes also increases fatigue load on the downstream turbines by $5\%$ to $15\%$, which can shorten their lifespans [43].

The wind farm control problem is challenging. Conventional model-based control strategies require tractable models of complex dynamic interactions between turbines, and suffer from the curse of dimensionality when the number of turbines increases. Moreover, optimal strategies differ significantly with modeling choices. Reinforcement Learning (RL) provides a model-free, data-based alternative, and recent work applying RL algorithms to wind farm control has yielded promising results (see e.g. [1]). Single agent approaches, where a single RL controller must learn a centralized policy, encounter scaling challenges [11], are slow to converge under dynamic conditions [25] and do not explore the graph structure of the problem induced by local perturbations. Several multi-agent RL approaches have been proposed to tackle this issue, relying on both centralized critics [9, 11, 31] and independent learning approaches [5, 22, 39]. Authors have published code relative to their specific applications [46, 45, 29], and [29] proposes a single-agent RL environment for power maximization in static simulations. There is to the best of our knowledge no open-source reinforcement learning environment for the general wind farm control problem.

In this article, we propose WFCRL, the first open suite of reinforcement learning environments for the wind farm control problem. WFCRL is highly customizable, allowing researchers to design and run their own environments for both centralized and multi-agent RL.

Wind turbines can be controlled in several ways. A turbine can adjust its *yaw* (defined as the angle between the rotor and the wind direction) to deflect its wake, increase its *pitch* (the angle between the turbine blades and the incoming wind) to decrease its wind energy production, or directly control the *torque* of its rotor. WFCRL makes it possible to control yaw, pitch or torque, and a schema of these different control variables can be found in Figure 1. WFCRL offers a large set of observations including local wind statistics, power production, and fatigue loads for each turbine. This makes it possible to consider different objective, including the maximization of the total production, the minimization of loads to reduce maintenance costs over the wind turbine life-cycle [23], or, as wind energy becomes a larger part of the energy mix, the tracking of power or frequency targets that allows operators to offer ancillary services for grid integration [26].

In WFCRL, interfaces with two state-of-the-art farm simulators are implemented : a static simulator FLORIS [13] and a dynamic simulator FAST.Farm [21]. Indeed, the choice of a static or dynamic model is particularly important: the overwhelming majority of proposed approaches are evaluated on static models, but it was shown in [40] that successful learning approaches under static conditions generally do not adapt to dynamic ones. However, online learning from scratch with dynamic simulators is often too slow, making transfer learning from static to dynamic simulators of great interest. From the broader literature on transfer learning and learning from simulators we know that it is challenging to train policies that can improve on previously learned behavior when deployed

on new environments with unseen dynamics [48, 15]. In spite of this problem, to the best of our knowledge, most approaches so far have been trained and evaluated on the same environment, and it is therefore not clear whether the policies learned with simulators are robust enough to be useful, or even safe, when deployed on real wind farms. With two simulators of different model-fidelity (referring to how closely the model represents the real system), WFCRL offers the possibility of designing transfer learning strategies between these simulators.

**Contributions of the paper**

- We introduce WFCRL, the **first open reinforcement learning** suite of environments for **wind farm control**. WFCRL is highly customizable, allowing researchers to design and run their own environments for both centralized and multi-agent RL. It includes a default suite of wind farm layouts to be used in benchmark cases.

- We interface all our wind farm layouts with **two different wind farm simulators**: a static simulator FLORIS [13] and a dynamic simulator FAST.Farm [21]. They can be used to **design transfer learning strategies**, with the goal to **learn robust policies that can adapt to unseen dynamics**.

- We include implementations of three state-of-the-art MARL algorithms, IPPO, MAPPO [47], and QMIX [34] adapted to our environments.

- We propose a benchmark example for **wind power maximization** with two wind condition scenarios. It takes into account the costs induced by wind turbine fatigue.

The paper is organized as follows. In Section 2, we introduce the WFCRL environment suite. First in Section 2.1 we introduce the simulators, the specifications of the simulated wind farms and turbines and the wind conditions scenarios we consider. We then lay out in Section 2.2 the cooperative MARL framework for the wind farm control problem, and finally detail the learning tasks and algorithms available with the suite in Section 2.3. In the second part Section 3, we illustrate the possibilities of the WFCRL environment suite by introducing a benchmark example: the maximization of total power production with fatigue-induced costs. In Section 3.1, we explicit the actions, observations and rewards used in this problem, then in Section 3.2, we present and discuss the results of the IPPO and MAPPO on our benchmark tasks. In Section 4, we discuss perspectives and limitations, and we conclude in Section 5

## 2   WFCRL environments suite

In this section, we present our WFCRL environments suite. We first present the simulators interfaced in WFCRL (FLORIS and FAST.Farm), several pre-defined layouts and wind condition scenarios. Note again that having two simulation environments with different model-fidelity offers the possibility of designing transfer learning strategies between simulation environments. Then, we describe briefly the MARL framework for the wind farm control problem. More precisely, we consider a wind farm with $M$ turbines, which operate in the same wind field and create turbulence that propagates across the farm. In our multi-agent environment, each turbine is considered an agent receiving local observations, and all cooperate to maximize a common objective. Note that WFCRL also has a single-agent RL environment, which uses global observations and actions.

### 2.1   The simulation environments

In WFCRL, users can choose one of the two state-of-the-art wind farm simulators (FLORIS or FAST.Farm), select a pre-defined wind farm layout or define a custom one, and choose one of the implemented wind conditions.

Various wind farm simulators can be used to evaluate wind farm control strategies [3]. The choice of the wind farm simulators included in WFRCL relies on three criteria: the trade-off between fidelity and computation time, the popularity of the simulators in the wind farm energy community, and open-source availability. We discuss this further in Appendix A, and introduce our simulation environments in the following paragraph.

| WFCRL environment | Real wind farm |
|---|---|
| Ablaincourt | Ablaincourt Energies onshore wind farm, Somme, France |
| Ormonde | Ormonde Offshore Wind Farm, Irish Sea, UK |
| WMR | Westermost Rough Wind Farm is an offshore wind farm, North Sea, UK |
| HornsRev1 | Horns Rev 1 Offshore Wind Farm, North Sea, Denmark |
| HornsRev2 | Horns Rev 2 Offshore Wind Farm, North Sea, Denmark |

Table 1: Correspondences between WFCRL environments and real wind farms.

**FLORIS environments**    The wind farm simulator FLORIS implements static wind farm models, which predict the locations of wake centers and velocities at each turbine in the steady state: the dynamic propagation of wakes are neglected. The yaws of all wind turbines can be controlled, and the power production of the wind farm is then a function of all yaw angles and the so-called free-stream wind conditions: wind measurements - e.g. velocity and direction - taken at the entrance of the farm. FLORIS has been released as an open-source Python software tool[1]. In WFCRL environments built on FLORIS, global and local states contain time-averaged, steady-state wind and production statistics for both global and local observations.

The models used by FLORIS do not compute any estimate of fatigues on wind turbines, and we propose to use local wind statistics to compute a proxy for load estimates indeed. We detail this when introducing our benchmark example in Section 3.1.

**FAST.Farm environments**    Unlike FLORIS, FAST.Farm is a dynamic simulator that produces time-dependent wind fields that take into account the dynamics of wake propagation [21]: wakes in wind farms tend to meander, and the wakes of different turbines interact and eventually merge as they propagate in the farms. One consequence is that under dynamic conditions there is a significant delay between the time agents take an action and the time this action finally impacts the turbines downstream.

FAST.Farm is built on wind turbine simulation tool OpenFAST [30] which computes an estimate of the strength of the bending moment on each turbine blades. This reflects the structural loads induced on turbine blades, and thus can be used to design rewards in RL problems to reduce or avoid physical damages to turbines.

FAST.Farm is coded in Fortran. To allow for integration with the large ecosystem libraries and RL research practices developed in Python, we implement an interface between the simulator and the Python wind farm environment via MPI communication channels. The details of the interfacing infrastructure are reported in Appendix B.

**Wind farm layouts**    Any custom layout - the arrangement of the wind turbines in the farm - can be used in WFCRL. We also propose several pre-defined wind farm layouts for use in benchmark cases. The coordinates of the wind turbines of 5 real wind farms with 7 to 91 wind turbines are obtained from [2]. A complete list of all correspondences between wind farms inspired by real environments and their locations is in Table 1, and a list of all available environments can be found in Appendix C. We also include in WFCRL several toy layouts, including a simple row of 3 turbines (the *Turb3Row1* layout) for validation purpose and the 32 turbines layout of the *FarmConners* benchmark [14]. A visual representation of the layouts can be found in Appendix H.

For all cases, we simulate instances of the NREL Reference 5MW wind turbines, whose specifications have been made public by the National Renewable Energy Laboratory (NREL) [20]. It has become standard reference for wind energy research and is used by the majority of proposed evaluations of RL methods [1].

**Wind condition scenarios**    For all environments, we distinguish three scenarios.

*Wind scenario I:* In this scenario, all trajectories in a given environment are run under the prevailing wind velocity and direction at the location.

---

[1]https://nrel.github.io/floris/

*Wind scenario II:* In this scenario, we let the wind farm be subject to variations in wind change, and sample new free-stream wind conditions $u_\infty, \phi_\infty$ at the beginning of each episode:

$$u_\infty \sim \mathcal{W}(\bar{u}, \lambda) \quad \phi_\infty \sim \mathcal{N}(\bar{\phi}, \sigma_\phi) \tag{1}$$

where $\mathcal{W}$ is a Weibull distribution modeling wind speed with shape $\lambda$ and scale $\bar{u}$, and $\mathcal{N}$ is a Normal distribution with $\bar{\phi}$ being the dominant wind direction for a given farm.

*Wind scenario III:* In this scenario, the wind farm is subjected to a an incoming wind that varies during a single episode. Any time series with wind speed and direction measurements can be used by WFCRL, and we provide a default time series of measurements collected on a real wind farm. A starting point in the time-series is randomly selected at the beginning of each new episode.

By default, at the beginning of each simulation, all wind turbines have the yaw angle zero. This corresponds to the so-called *greedy* case, the strategy that would allow each of them to maximize its production in un-waked conditions.

## 2.2  The MARL framework for the wind farm control problem

A Decentralized Partially Observable Markov Decision Process (Dec-POMDP) with $M$ interacting agents is a tuple $\{M, S, O, A, P, o^1, \ldots, o^M, r\}$. $S$ is the full state space of the system, while for any $i \in \{1, \ldots, M\}$, $O_i$ is the observation space of the $i$th agent with $O = \times_i^M O_i$. $A_i$ is the local action space of the agent, and the global action space is the product of all local action spaces $A = \times_i^M A_i$. At each iteration, all agents observe their local information, chose an action and receive a reward $r : S \times A \times S \to \mathbb{R}$. The system then moves to a new state, which is sampled from the transition kernel $P : S \times A \times S \to [0, 1]$. $P$ gives the probability of transition from a state $s \in S$ to $s' \in S$ when agents have taken global action $a \in A$. The probability for the $i$th agent to observe $o_i$ is then defined by local observation function $o^i : S \times A \times O_i \to [0, 1]$, and the history of all past observations is denoted $h_i$. We call $\pi_1, \ldots, \pi_M$ the policies followed by each agent, where $\pi_i(a_i|h_i)$, defines the probability for agent $i$ to chose action $a_i$ after observing $h_i$. The corresponding global policy $\pi = (\pi_1, \ldots, \pi_M)$ simply concatenates the outputs of all local policies.

**Objective**  The MARL problem is to find a policy $\pi^*$ that maximizes the expectation of the discounted sum of rewards collected over a finite or infinite sequence of time-steps

$$\max_\pi \mathbb{E}_{s_0, a_0, s_1, \ldots} [J], \quad J := \sum_{k=0}^T \beta^k r_k \tag{2}$$

with $0 < \beta < 1$ the discount factor and $T$ the number of steps in the environment, or the length of an episode. For the wind farm control problem, possible rewards include the total production of the farm or a distance to a target production. As fatigue load measurements are also available, rewards can be designed to encourage actions that preserve the turbine structure. Note that knowledge of the farm layout and incoming wind direction can also be exploited to represent wake interactions between wind turbines as a time-varying graph (see Appendix G): this approach can motivate the design of local reward functions for decentralized learning RL algorithms with communication. Moreover, empirical successes applying RL to wind farm control have often relied on creative reward shaping [1, 10]. To support these two efforts, WFCRL implements a `RewardShaper` class that allows easy design of custom reward functions, and can be used to train both centralized and decentralized learning algorithms.

**State and Observation**  As the production of each turbine is a function of the local wind conditions at its rotor, a Markovian description of the full state of the system should contain the whole wind velocity field of the entire farm. This is impossible to know in practice. We rather assume that local measurements of wind speed and direction are available at each wind turbine, and that an estimate of the free-stream wind speed and direction can be accessed, but might not necessarily be sent to the turbines in real time. Our environments therefore distinguish between the local observations $o_i$ for each i $\in \{1, \ldots, M\}$ and a global observation $o_g$. Each $o_i = (u_i, \phi_i, \theta_i)$ contains a local measure of the wind velocity $u_i$ and direction $\phi_i$, as well as the last target sent to each actuator $\theta_i$. On FLORIS environments, $\theta_i$ is always equal to the current value of the actuators. This is not the case on FAST.Farm, for which an inner control loop at the level of the actuators adds a response delay. The global observation $o_g = (o_i, \ldots, o_M, u_\infty, \phi_\infty)$ contains the concatenation of all local states, as well

|  | FLORIS | FAST.Farm |
|---|---|---|
| Local Observations $o_i$ | $u_i, \phi_i$ (steady-state), $y_i$ | $u_i, \phi_i$ (time-dependent), $y_i, p_i, \tau_i$ |
| Global Observations $o_g$ | $o_1, \ldots, o_M, u_\infty, \phi_\infty$ | |
| Actions | $\Delta y_i$ | $\Delta y_i, \Delta p_i, \Delta \tau_i$ |

Table 2: Observations (global and local) and actions available for an agent $i$ in FLORIS and FAST.Farm environments. $y_i, p_i, \tau_i$ refer respectively to the yaw, pitch and torque of the turbine.

as the free-stream measure of the wind $u_\infty, \phi_\infty$. Table 2 summarizes all observations and actions available with the two simulators.

**Actions**   WFCRL offers several ways to control wind turbines: the *yaw*, the *pitch* or *torque*. Yaw control is available on the FLORIS environments, and all three can be controlled on FAST.Farm environments. The yaw is the angle between a wind turbine's rotor and the wind direction: turbines facing the wind have a yaw of $0°$ which maximizes their individual power output. Increasing the yaw can deflect the wake away from downstream turbines, which may increase the total production of the wind farm. The pitch is the angle of the attack of the rotor blades with respect to the incoming wind, while the torque of the turbine's rotor directly controls the rotation speed. Increasing the blade pitch or decreasing the torque target both decrease the fraction of the power in the wind extracted by the turbine, and therefore decrease the turbulence in its wake. To reflect the fact that the actuation rate of the wind turbines is limited by physical constraints, we conceive actions as increases or decreases in the actuator target value rather than absolute values, with the limits being implemented by the upper and lower bounds of a continuous action space.

## 2.3   Learning in WFCRL

All environments are implemented with standard RL and MARL Python interfaces Gymnasium [7] and PettingZoo [42]. The source code of WFCRL is open-sourced under the Apache-2.0 license and publicly released at `www.github.com/ifpen/wfcrl-env`.

### 2.3.1   Online Learning

Environments implemented on both FLORIS and FAST.Farm can be used in an episodic learning approach. This is the traditional setting of the RL problem, and we will refer to it as the *Online Learning* Task. In Wind scenario I and III, we look at the evolution of the sum of rewards collected over an episode. In Wind scenario II, where a different set of wind conditions is sampled at each episode, we evaluate the policies on a predefined set of wind conditions and use a weighted average as the final score. This gives us our evaluation score:

$$\texttt{score}(\pi_1, \ldots, \pi_M) = \sum_{j=1}^{n_w} \rho_j \sum_{k=0}^{T} r_k \tag{3}$$

where $T$ is the length of the episode, $n_w$ is the number of wind conditions considered and the $\rho_1, \ldots, \rho_{n_w}$ are the weights on each conditions, with for all $j$, $0 < \rho_j < 1$ and $\sum_{j}^{n_w} \rho_j = 1$. The wind conditions distributions on which policies are evaluated need not be identical to the one from which conditions were sampled during training.

### 2.3.2   Transfer

Exploration on real wind farms is costly: as prototype models are typically not available for large wind farms, adjusting to the real dynamics of the system will require exploring in real time on an operating wind farm. Every move of exploring in a suboptimal direction is a cost for the farm operator. Learning efficient policies offline that can quickly adapt to the real system is therefore critical. Since the dynamic FAST.Farm simulator is considered a higher fidelity version of the static simulator FLORIS, we propose to use the former as a proxy of a real wind farm to evaluate the robustness of policies learned on the latter, and their ability to adjust to the real dynamics of a farm. We will refer to this as the *Transfer* Task.

### 2.3.3   Algorithms

We focus on two state-of-the-art algorithms IPPO (Independent PPO) and MAPPO (Multi-Agent PPO) introduced in [8, 47]. Both are based on the on-policy actor-critic PPO algorithm [35], and support

continuous action spaces. Following an approach called independent learning, IPPO builds on PPO by allowing every agent to run a PPO algorithm in parallel. MAPPO, on the other hand, maintains both $M$ agent policies taking actions based on local information and a shared critic, which estimates the value of a global observation. The choice of the global observation fed to the critic is an important factor influencing the performance of the algorithm [47]. We follow the recommendations of [47] to adapt PPO to the multi-agent case. They suggest to include both local and global observation features to the value function input. We therefore feed to our shared critic network the full global observation introduced in Section 2.2, that is both the of concatenation of all local observations and the free-stream wind velocity.

In [47], it was found that PPO-based methods perform very well when extended to cooperative multi-agent tasks, outperforming algorithms specifically designed for cooperative problems like QMIX [34]. We implement both QMIX and independent Deep Q-network (DQN) [27] baselines, where all agents run DQN algorithms in parallel using a discrete action space. As QMIX uses a recurrent neural network, we implement independent DQN baselines with both fully connected (IDQN) and recurrent (IDRQN) neural networks. The same global observation is fed to the central MAPPO critic and the mixing network of QMIX.

For implementation, we adapt the CleanRL [2] [18] baseline implementations of PPO and DQN to our multi-agent Petting Zoo environments. Although our analysis for the benchmark case considered in Section 3 focuses on PPO-based algorithms on continuous action spaces, additional results for baselines on discrete action spaces are reported in Appendix F.

## 3 Benchmark example: the maximization of the total power production

We consider the problem of finding the optimal yaws to maximize the total power production under a set of wind conditions, and taking into account the costs induced by turbine fatigue load. This problem is known as the *wake steering* problem, and is an active area of research in the wind energy literature [16, 17].

### 3.1 Problem formulation

**Actions and observations**   Local observations include the local yaw and local wind statistics. The concatenation of all local observations along with free-stream wind statistics in the global observation is as described in Section 2.2. Recall that actions are defined as increase or decrease in the actuator target value. In this problem, all agents control their yaws, and we define the continuous action space $[-5, 5]$, defining changes in yaw angle expressed in degrees. To constraint the load on the turbines caused by the control strategies and reduce its impact on the lifetime of the turbines, the time each turbine spends actuating is limited. We choose the upper bound of $10\%$ of the time, which is the same upper bound value discussed in [33]. At every iteration, the time needed to change the state of the actuator is computed, and any action violating this condition is not allowed.

**Rewards**   At each iteration $k$, all agents receive a reward $r_k^P$ which is the currently measured production of the wind farm in kW divided by the number of agents and normalized by the free-stream wind velocity:

$$r_k^P = \frac{1}{M} \sum_i^M \frac{\hat{P}_k^i}{(u_{\infty,k})^3} \tag{4}$$

where $\hat{P}_k^i$ is the measured power production and $u_{\infty,k}$ the free-stream wind velocity at time-step $k$. To discourage agents from taking risky policies damaging the turbines, we also return a load penalty $r_k^L$ which increases with the sum of loads on all the turbine blades.

FLORIS does not provide estimates of the loads on structures. Instead, we evaluate the impact of actuations on loads with a proxy based on local estimates of turbulence and velocities on the surface of the rotor planes. Our proxy takes into account 2 factors increasing stress on wind turbine structures as noted in [41]: first, the turbulence of the wind and second, the variation of velocities on the turbine

---

[2]https://github.com/vwxyzjn/cleanrl

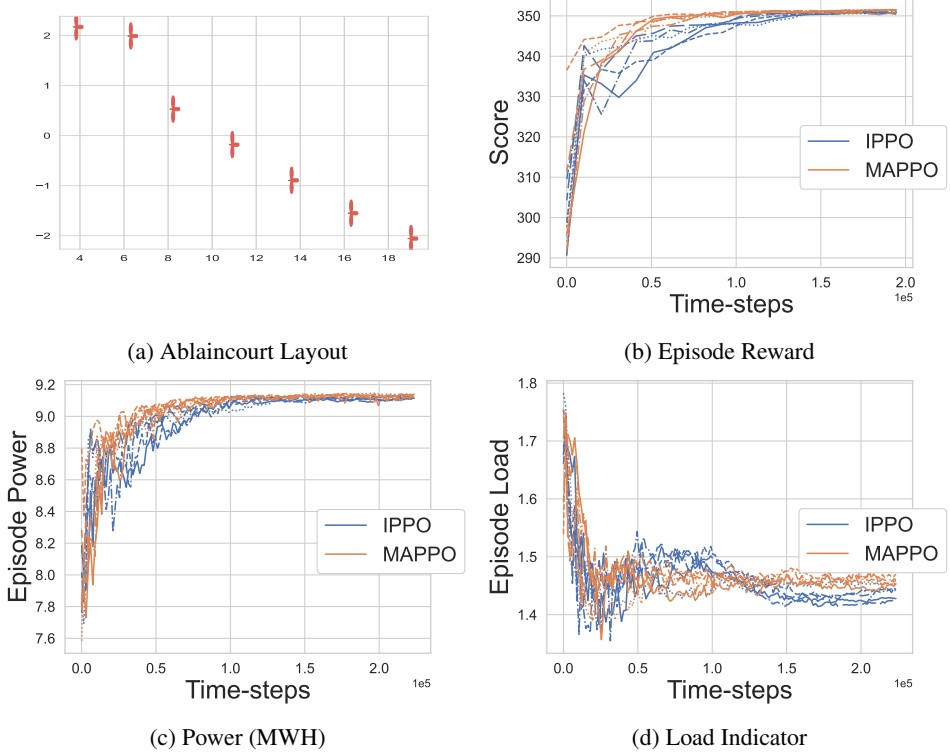

(a) Ablaincourt Layout

(b) Episode Reward

(c) Power (MWH)

(d) Load Indicator

Figure 2: The evolution of total reward (b), power output (c) and load penalties (d) accumulated over an episode (with $T$=150) on the *Ablaincourt* environment, simulated with FLORIS. A visual representation of the layout is in (a), where the coordinates are in wind turbine diameters. During training, policies are evaluated every 5 training steps with deterministic policies. The curves are plotted for all 5 seeds.

rotor. We therefore define the load penalty in FLORIS environments as

$$r_{k,S}^{L} = \frac{1}{M} \sum_{i}^{M} \left( \sum_{j}^{9} TI_k[x_{i,j}, y_{i,j}] + \sigma(u_k) + \sigma(v_k) + \sigma(w_k) \right) \quad (5)$$

where $TI_k$ is the turbulence field at time-step $k$, $u_k$, $v_k$ and $w_k$ are respectively the $x$, $y$ and $z$ components of the velocity field at time-step $k$, and the $x_{i,j}$ define the coordinates of the $9 \times M$ grid points at which these values are computed for the $M$ rotor planes. $\sigma$ denotes the standard-deviation.

For FAST.Farm, we use the estimates of the the blades' bending moment strength as a proxy for the structural loads induced on the turbines, and define the load penalty as

$$r_{k,D}^{L} = \frac{1}{M} \sum_{i}^{M} \left( \sum_{j}^{3} |Mop_k[i,j]| + \sum_{j}^{3} |Mip_k[i,j]| \right) \quad (6)$$

where $Mop_k$ is the $M \times 3$ matrix of out-of-plane bending moments for the 3 blades of every turbine at time-step $k$, and $Mip_k$ is the corresponding matrix of in-plane bending moments.

Both rewards are common to all turbines, and all must therefore maximize (2) with $r_k = (r_k^P - \alpha r_k^L)$, where $\alpha$ is a weighting parameter. The load penalty indicator returned by the environment is downscaled so that $r_k^P$ and $r_k^L$ are of similar magnitudes. By default, the value of $\alpha$ is 1, but greater or less importance can be given to turbine safety by changing $\alpha$.

**Wind conditions** We focus here on Wind Scenarios I and II. To evaluate the algorithms on Scenario II with score (3), we need to define weights $\rho_j$. We use data acquired during the SmartEole project at the location of the Ablaincourt wind farm [12]. It consists of estimates of free-stream wind direction

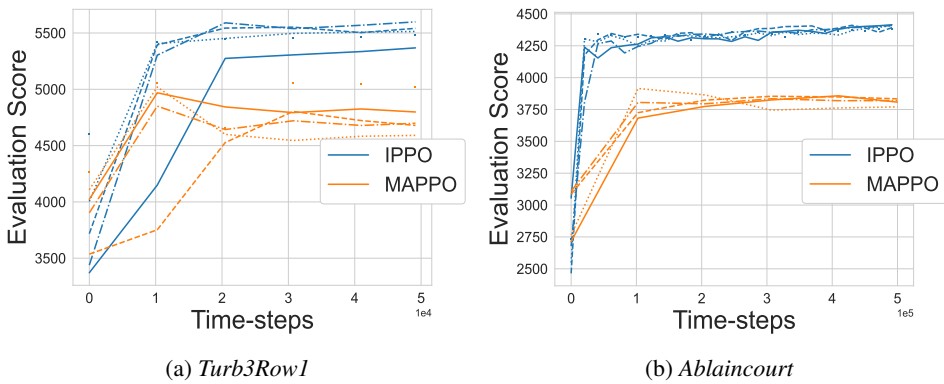

(a) *Turb3Row1*                                    (b) *Ablaincourt*

Figure 3: Evolution of the evaluation score, defined in (3), during the training of IPPO and MAPPO on the two environments *Turb3Row1* (left) and *Ablaincourt* (right).

and velocity computed from measures taken during a 3 months field campaign every 10 min. Since in real conditions wind velocity and wind direction are correlated, we compute the bi-dimensional histogram for the two variables, taking 5 bins for each dimension. We obtain a set of 25 wind condition rectangles. The wind conditions $w_1, \ldots, w_j$ are the center of each rectangle, and the corresponding weights $\rho_j$ are defined as the frequencies at which wind conditions in the time series appeared in the rectangle.

### 3.2   Results

We apply algorithms IPPO and MAPPO, whose implementations are both available in WFCRL, to our benchmark example. Both are trained on the two wind scenarios I and II described in Section 2.1, on the static simulator FLORIS. For the Scenario I, the score is the reward obtained on a single policy rollout of 150 steps in the environment, and policies are updated after 2048 steps in the environment. Learned deterministic policies are evaluated every 5 training steps. Results for the *Ablaincourt* layout are given in Fig. 2. For the Scenario II, the score is the one defined in (3), with $T = 2048$. The training curves for the the *Ablaincourt* and the *Turb3Row1* layouts are illustrated in Fig. 3.

At the end of training, we evaluate all algorithms on both scenarios, as well as on FAST.Farm environments for Scenario I. On *Turb3Row1*, IPPO has the best performance for all evaluation tasks, while on *Ablaincourt*, MAPPO performs better for 2 evaluation tasks out of 3. This confirms the good empirical results of IPPO on cooperative tasks observed in the literature [8], and suggests that MAPPO's shared critic becomes more beneficial as the number of agents increases. Although IPPO and MAPPO perform similarly on Scenario I, the gap in favor of IPPO increases on Scenario II, showing MAPPO to be less efficient at adapting policies to diverse wind conditions.

As expected, the best evaluation performance for each wind scenario on FLORIS is provided by the algorithms trained on this scenario. Yet on FAST.Farm, although our evaluation task is solely under Scenario I (constant wind), its noisier wind observations pose a challenge to IPPO policies trained on Scenario I. As local wind observations are now perturbed by time-dependent turbulences created by other agents, information sharing (MAPPO) or exposition to a variety of wind conditions (Scenario II) during training becomes more useful.

A table detailing all evaluation scores at convergence is available in Appendix F and hyper-parameters are given in Appendix D. The code to reproduce all experiments is available at `www.github.com/ifpen/wfcrl-benchmark`.

To illustrate the *Transfer* case, we then fine-tune the learned IPPO policies on a *Turb3Row1* on $40k$ steps (1 day in simulated time) in the corresponding FAST.Farm environment. We report in Appendix F.1 the evolution of average power output and load, and compare it to a naive deployment of strategies learned online. Our results illustrate the difficulty of adapting learned policies to unseen dynamics.

## 4 Limitations

As noted in Section 2, the choice of the simulators included in WFRCL has been made on the criteria of fidelity, computation cost, popularity and availability in open source. Both FLORIS and FAST.Farm are developed and actively maintained by the US-based National Renewable Energy Laboratory [3], and have a large user base among wind energy researchers. FAST.Farm was explicitly designed to provide good fidelity at a limited computation cost [21]. Despite this, dynamic wind farm simulators remain slow. The development and open-sourcing of faster dynamic simulators will be critical. Machine-learning accelerated simulators could be an important step in that direction. Moreover, although FAST.Farm has been extensively validated against both real wind farm data and high-fidelity simulations (see Appendix A), there has been to the best of our knowledge no explicit investigation of the transfer of yaw optimization results from FAST.Farm simulations to a real wind farm.

## 5 Conclusion

We have introduced WFCRL, the first open reinforcement learning suite of environments for wind farm control. WFCRL is highly customizable, allowing researchers to design and run their own environments for both centralized and multi-agent RL. It is interfaced with two different wind farm simulators: a static simulator FLORIS and a dynamic simulator FAST.Farm. They can be used to design transfer learning strategies with the goal to learn robust policies that can adapt to unseen dynamics. We have proposed a benchmark example for wind power maximization with two wind condition scenarios that take into account the costs induced by wind turbine fatigue. We hope that WFCRL will help building a bridge between the RL and wind energy research communities.

---

[3]https://www.nrel.gov/

# 6 Acknowledgments

Part of this work is carried out in the framework of the AI-NRGY project, funded by France 2030 (Grant No: ANR-22-PETA-0004).

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

Figure 4: Wind velocity field for the simulation of our 3-turbines layout on the 2 simulators: FLORIS and FAST.Farm.

## A    Difference between FLORIS and FAST.Farm

Wind farm models serve two main purposes, in the broader literature as in WFCRL. First, when experiments on real wind farms or tunnel experiments on scaled farms are not possible, they are the only way to evaluate and compare control strategies by predicting their impact on the total power output of a farm. For that purpose, the value of models lie in their accuracy, and the best results are achieved by complex dynamic models involving costly computations. Secondly, a model of the farm can be used to estimate an optimal command. Here, accuracy must be balanced by tractability, and a constraint on computation time arises for real-time optimization. Of course, evaluating the command on the model used to derive it will likely overestimate its performance: it should rather be evaluated on an other, higher fidelity model which will serve as a substitute for the real farm.

Static models estimate the time-averaged features of the wind flow while ignoring the dynamics of short-term effects, including wake propagation time and wake meandering. They rely on the design of an analytical solution to predict wind speed deficit at a downwind turbine with respect to an upstream turbine [3]. This gives them the advantage of a very low computation time, as they usually return a solution instantaneously. The wind farm simulation software FLORIS (NREL 2021), created and maintained by the National Renewable Energy Laboratory (NREL), proposes a variety of such models in a single Python framework, and has become a reference for wind farm control engineering. These parametric models combine several components to estimate the effects of turbine yaws on both the redirection of the wake behind the turbine and the velocity in the wake. An example of such a simulation can be found on Figure 4a.

At higher accuracy and higher computational complexity is FAST.Farm [21]. It relies on OpenFAST (NREL 2022) to model the dynamics of each individual turbine, but considers additional physics to account for farm-wide ambient wind, as well as wake deficits, propagation dynamics and interactions between different wakes. It supports the implementation of controllers tracking a received yaw reference for each turbine. Figure 4b provides an example of these realistic wind fields.

FAST.Farm has been shown to be of similar accuracy with high-fidelity large-eddy simulations with much less computational expense. It has been validated against both real wind farm data [36, 24] and Large Eddy Simulations [19, 37]. Power predictions match real measurements within 2-7% error on average [36], although the error can reach 18% for downstream turbines under low free-stream wind speeds. Load predictions are within 10% of groundtruth values, with errors reaching 25% for certain wind directions [37]. Predictions always respect trends in power and load changes: under the same wind conditions, yaws that lead to increases in power output in simulations also lead to increases in real wind farms.

# B   Details of the FAST.Farm interface

The Python-FAST.Farm interfacing tool relies on two interfaces with RECEIVE and SEND functions. Following [38] in which the authors designed an interface between FAST.Farm and Matlab based on the MPI communication protocol, we rely on an MPI communication channel between the Python and FAST.Farm processes. We choose to let the Python process spawn a new child process to launch the FAST.Farm simulation in the background, allowing the user to only interface with Python. The architecture of the interfacing tool is illustrated in Figure 5.

At every iteration, the FAST.Farm interface retrieves 12 measures per turbine:

- 2 wind measurements: wind velocity and direction at the entrance of the farm. The wind direction is estimated by subtracting the yaw estimation error from the current yaw measure.
- The current output power of the turbine
- The yaw of the turbine
- The pitch of the turbine
- The torque of the turbine
- 6 measures of blade loads: the out-of-plane bending moment estimate for each blade, and the in-plane bending moment estimate on each blade

and sends the 3 control targets - yaw, pitch, torque - to each local turbine controller. Other actuators that are not controlled by the RL algorithm are controlled by the default naive FAST.Farm controllers.

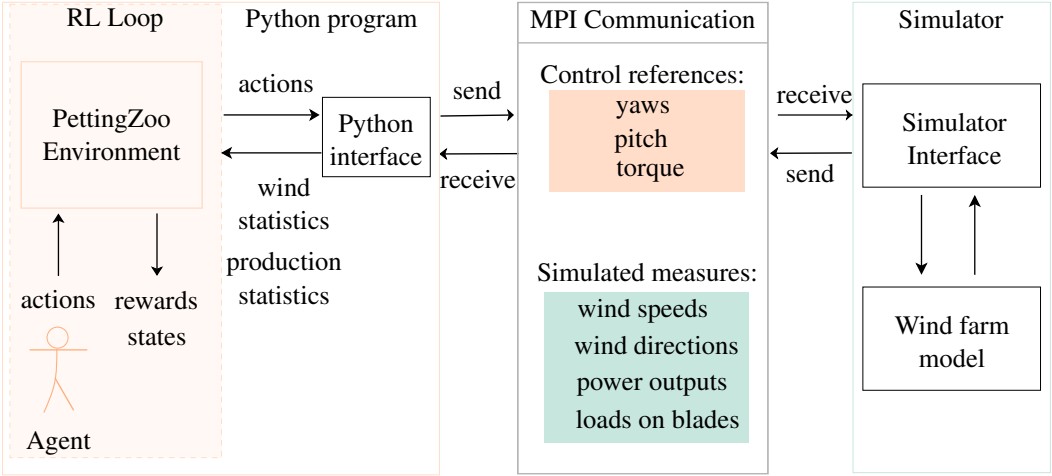

Figure 5: Schema: interfacing infrastructure between FAST.Farm and Python

Free-stream wind speed and direction are estimated as the wind measurements at the wind turbine with the highest wind speed. Since inflow wind can be turbulent, all estimates are averaged over a period of time defined by the *buffer_window* parameter.

# C Characteristics of all environments

|  | **Centralized Control** | **Decentralized Control** |
|---|---|---|
| **FLORIS** | *LayoutName_Floris* | *Dec_LayoutName_Floris* |
| **FAST.Farm** | *LayoutName_Fastfarm* | *Dec_LayoutName_Fastfarm* |

Table 3: Creating environment IDs: Prefix, Root, Suffix

For preregistered layouts, every environment is characterized by a tuple of 3 options, and every environment ID is a combination of the corresponding 3 parts: a prefix, a root, and a suffix.

- The choice to formalize it as a centralized or decentralized control problem. Environments with centralized control are *Gymnasium* environments and expect global actions, i.e. vectors concatenating all actions, and have no prefix. Environments with decentralized control are *PettingZoo* environments and expect local actions sent by each agent. They have the prefix *Dec*.

- The choice of the layout, i.e. the arrangement of wind turbines in the field. A list of all layouts is given in Table 4, and a visual overview of them in Appendix H. The name of the layout is the root of the environment ID.

- The choice of a simulator. Two simulators are for now implemented in WFCRL: the static FLORIS and the dynamic FAST.Farm. The corresponding suffix *Floris* or *Fastfarm* is appended to the environment ID.

| **Layout Name** | **# Agents** | **Description** |
|---|---|---|
| Ablaincourt | 7 | Inspired by layout of the Ablaincourt farm in France, (Duc et al, 2019) |
| Turb16_TCRWP | 16 | Layout of the Total Control Reference Wind Power Plant (TC RWP) (the first 16 turbines) |
| Turb6_Row2 | 6 | Custom case - 2 rows of 6 turbine |
| Turb16_Row5 | 16 | Layout of the first 16 turbines in the CL-Windcon project as implemented in WFSim |
| Turb32_Row5 | 32 | Layout of the farm used in the CL-Windcon project as implemented in WFSim |
| TurbX_Row1 for X in [1, 12] | X | Procedurally generated single row layout with X turbines, spaced by 4D with the D the diameter of the turbine. |
| Ormonde | 30 | Layout of the Ormonde Offshore Wind Farm |
| WMR | 35 | Layout of the Westermost Rough Offshore Wind Farm |
| HornsRev1 | 80 | Layout of the Horns Rev 1 Offshore Wind Farm |
| HornsRev2 | 91 | Layout of the Horns Rev 2 Offshore Wind Farm |

Table 4: Preregistered layouts: name, number of agents, and description

# D Environment and Training procedure details

The source code for the WFCRL package is open-sourced under the license Apache v2, and publicly released here `www.github.com/ifpen/wfcrl-env`, along with notebook tutorials and documentation. An example of code snippet allowing the creation of the FLORIS Ablaincourt environment with decentralized control is given below:

```
from wfcrl import environments as envs
env = envs.make("Dec_Ablaincourt_Floris")
```

The code to reproduce all experiments is available here `www.github.com/ifpen/wfcrl-benchmark`. Our algorithm implementations use *PettingZoo*'s Agent Environment Cycle [42] interaction logic. Our multi-agent environments also support standard *Gymnasium*-like RL control loops via the *PettingZoo*'s `aec_to_parallel` function.

We report in Table 5 the hyper-parameters used for IPPO and MAPPO, and in Table 6 the hyper-parameters used for QMIX and IDRQN. Recurrent Q-networks in QMIX and IDRQN take as input both the current local observation and the last action taken by the agent. All algorithms are trained on episodes of length $T = 150$ (*Sc. 1*) or $T = 2048$ (*Sc. 2*) with $\beta = 0.99$. Evaluation is done on episodes of length $T = 150$.

| Parameter | Value |
|---|---|
| Learning rate | $0.0003 \rightarrow 0$ (linear annealing) |
| $\beta$ | 0.99 |
| GAE $\lambda$ | 0.95 |
| # minibatches | 32 |
| # epochs | 10 |
| Normalize advantages | True |
| Clip coefficient | 0.2 |
| Value loss coefficient | 0.5 |
| Maximum gradient norm | 0.5 |
| Hidden layers | (64, 64) |
| # steps between updates | 2048 |
| Batch size | 2048 |
| Minibatch size | 64 |

Table 5: PPO Experiment hyperparameters (used in IPPO and MAPPO)

| Parameter | Value |
|---|---|
| *Shared parameters (Q-networks)* | |
| Learning rate | 0.0005 |
| $\beta$ | 0.99 |
| Hidden layers | 64 |
| # layers | 2 |
| Batch size (episodes) | 32 |
| Buffer size (episodes) | 200 |
| Exploration $\epsilon$ | $1 \rightarrow 0.05$ |
| Target network frequency (episodes) | 25 |
| $i$th Q-Network input | $(o_t^i, a_{t-1}^i)$ |
| *QMIX Hypernetwork Parameters* | |
| Hidden layer dim | 32 |
| # layers | 2 |

Table 6: QMIX and IDRQN (Independent DQN with deep recurrent network) experiment hyperparameters

The experiments were run on 3 different computers. The first computer, which has no GPU and a Intel Xeon Gold 6240Y processor, was used to train IPPO and MAPPO on *Wind Scenario I* for 1 week of compute. On the second computer, an internal cluster with a GPU Quadro RTX 6000 24Go, 2 week of compute was used to to train experiments of *Wind Scenario II*. The last computer which has a Intel Xeon Gold 6240Y processor was used for training models during 3 days of compute on *Wind Scenario I*, and for evaluation purposes.

# E  Score: wind rose and weights

In this section we illustrate the use of wind statistics from the SMARTEOLE dataset to extract wind conditions weights $\rho$ of the evaluation score (3). In Figure 6a, we report the distribution of wind velocity and direction in the SMARTEOLE dataset. In Figure 6b, we show the corresponding extracted weights $\rho$ for the 25 corresponding wind conditions.

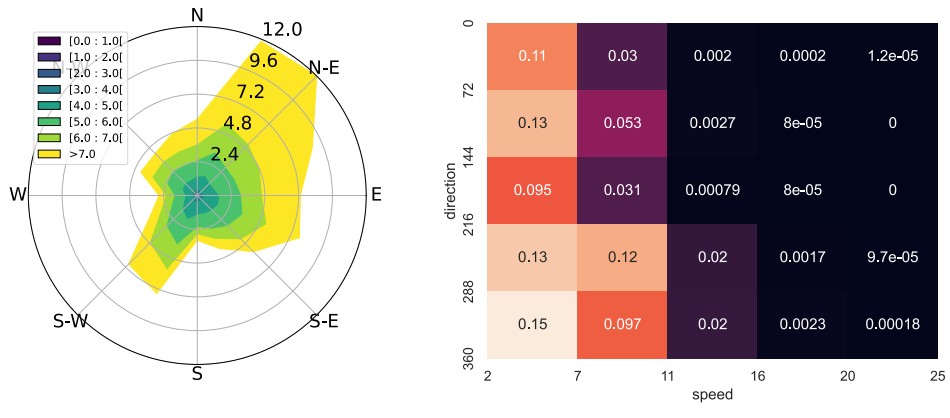

(a) Wind conditions in SMARTEOLE        (b) $\rho_i$ extracted from SMARTEOLE

Figure 6: Extraction of the $\rho_i$ weights from the SMARTEOLE dataset. The empirical distribution of wind speed and direction in the data represented as a windrose is in (a), and the corresponding extracted weights $\rho_i$ given to each of the 25 wind conditions are in (b).

# F   More benchmark results

## F.1   Evaluation and transfer on FAST.Farm

The literature on transferring wind farm control policies from lower to higher fidelity simulators is scarce. In [28], a transfer method is proposed that relies on a complex combination of optimization in static models, supervised learning of policies, policy evaluation and problem-specific reward engineering. The development of simpler transfer solutions that are less problem-dependent will be necessary to progress towards the deployment of RL policies on real wind farms. Yet as we show here, a naive fine-tuning approach might not be enough. On this task, we simulate a 900 steps episode of the *Turb3Row1* layout on FAST.Farm (environment *Dec_Turb3_Row1_Fastfarm*).

For the *Transfer* task, we pursue the training in the new environments, and report the evolution of the power and load compared to the *Eval* case in Figure 7 for the agents trained under IPPO. We simulate a day of training on FAST.Farm, corresponding to 28800 steps in the environment. During

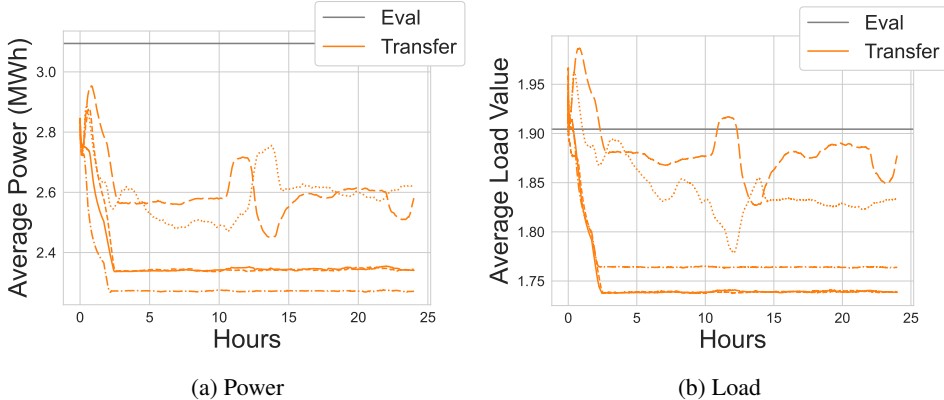

(a) Power                    (b) Load

Figure 7: Evaluation and transfer on FAST.Farm: evolution of power (left) and load (right) on the *Dec_Turb3_Row1_Fastfarm* environment. Results are reported for 5 seeds.

the evaluation tasks, policies learned on FLORIS (IPPO, *Sc. I*) deployed on the dynamic simulator achieve an increase of 15% in power production over the baseline. We know from existing literature that on *Turb3Row1*, there exists a policy that reaches an increase of 21% [28]. The difference in performance suggests that we could benefit from further fine-tuning these policies on the dynamic simulation. However, the simple transfer learning strategy pursued during the *Transfer* task degrades the performance of the policies when learning online, reaching an average of 0% increase over the greedy baseline at the end of the experiments. This illustrates the challenge of designing robust methods to bridge the gap between simple simulators and complex real world dynamics.

## F.2   More training results

In this section we report more benchmark results. The training curves of IPPO and MAPPO (resp. QMIX and IDRQN) under *Wind Scenario I* on the *Turb3Row1* layout are in Figure 8 (resp Figure 9). As the latter, value-based algorithms require a discretization of the action space, we report their results separately. We use an action space of 3 actions: increase actuation target value, decrease it, or do nothing. Each change in actuation leads to a change of 5° in the actuation space. Table 7 summarizes the evaluation scores, reporting mean and standard deviation on the 5 seeds.

We also report for the evaluated models on *Sc. I* the decomposition of the episode reward in power output (Table 8) and load penalty (Table 9). Table 8 is equivalent to the total energy produced during the episode, while Table 9 can be interpreted as an indication of accumulated damage. An example of a the rollout of a policies learned with IPPO on Scenario I is given in Figure 10 for *Turb3Row1* and *Ablaincourt*. In both layouts, the wake of the last turbine in the row has no impact on the production of the others. Its rotor is therefore maintained facing the wind, as in the greedy solution (yaw of 0°). On *Turb3Row1*, where a row of 3 turbines are perfectly aligned with the wind direction, learned policies alternatively lead the yaws of the first turbines to around +30° or −30°. On *Ablaincourt* on the other hand, learned policies always converge towards negative yaws.

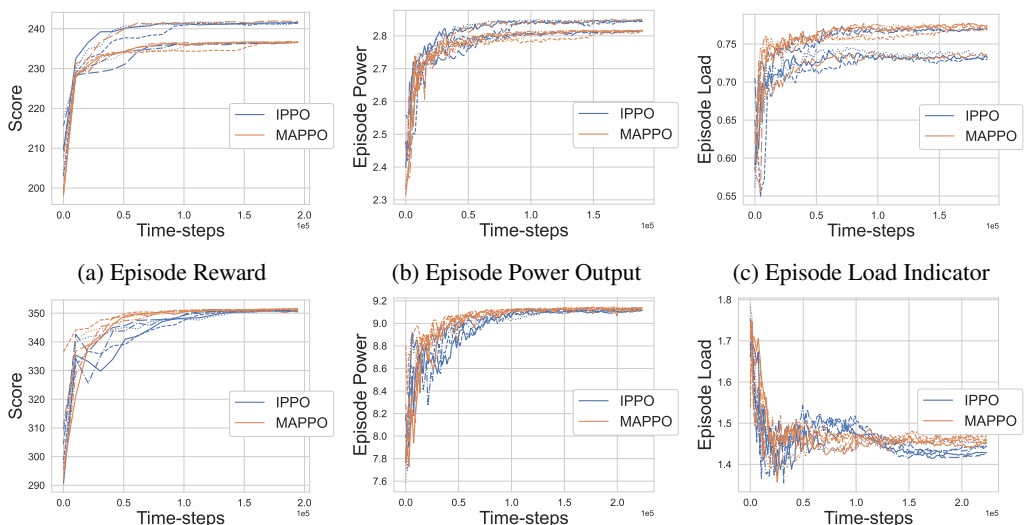

Figure 8: Evolution of episode reward (a), power output (b) and load indicator (c) on the layout *Turb3Row1* (top) and *Ablaincourt* (down) FLORIS environments for actor-critic algorithms IPPO and MAPPO during training. Deterministic policies are evaluated every 5 training steps.

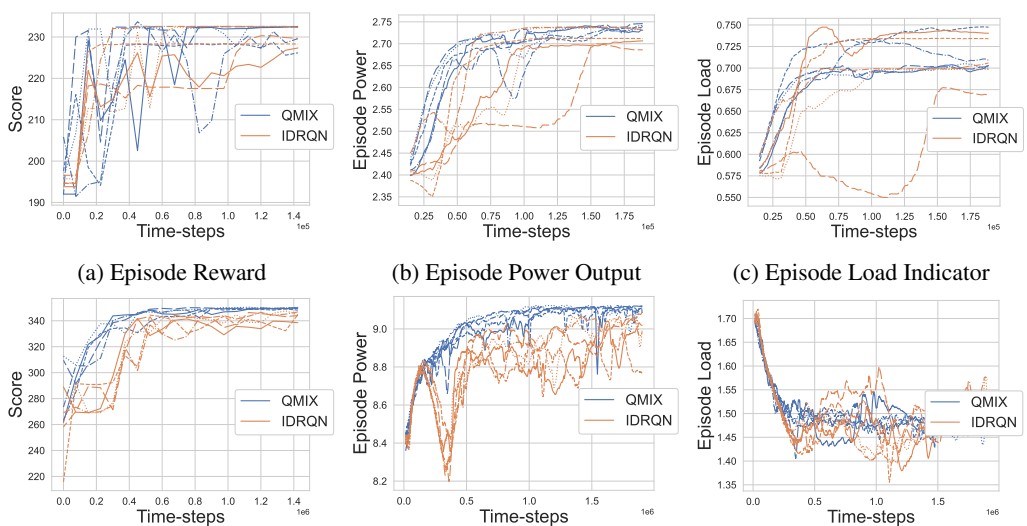

Figure 9: Evolution of episode reward (a), power output (b) and load indicator (c) on *Turb3Row1* (top) and *Ablaincourt* (down) FLORIS environments during training. Value-based approaches QMIX and IDRQN. Deterministic policies are evaluated every 50 episodes.

| | Turb3Row1 | | | Ablaincourt | | |
| | FLORIS | | FAST.Farm | FLORIS | | FAST.Farm |
| | Sc. 1 | Sc. 2 | Sc. 1 | Sc. 1 | Sc. 2 | Sc. 1 |
|---|---|---|---|---|---|---|
| | | | Training on Sc. 1 | | | |
| IPPO | **239.5** ± 2.4 | 354.8 ± 14.4 | 217.4 ± 1.8 | 351.0 ± 0.3 | 329.9 ± 1.4 | 327.4 ± 1.6 |
| MAPPO | 237.7 ± 2.1 | 345.1 ± 12.6 | 215.3 ± 1.4 | **351.7** ± 0.1 | 327.6 ± 1.4 | **330.2** ± 0.1 |
| IDQN | 202.6 ± 21.5 | 278.8 ± 31.8 | 173.5 ± 28.7 | 263.3 ± 11.2 | 231.4 ± 7.9 | 189.3 ± 10.5 |
| IDRQN | 210.6 ± 20.7 | 295.1 ± 32.2 | 184.8 ± 28.7 | 257.1 ± 2.5 | 240.8 ± 3.5 | 183.0 ± 0.6 |
| QMIX | 193.5 ± 17.0 | 273.1 ± 26.3 | 161.0 ± 23.4 | 254.8 ± 0.7 | 230.8 ± 11.1 | 181.7 ± 0.2 |
| | | | Training on Sc. 2 | | | |
| IPPO | 203.7 ± 4.3 | **406.0** ± 10.5 | **218.7** ± 0.5 | 321.8 ± 7.5 | **354.6** ± 4.1 | 317.3 ± 4.1 |
| MAPPO | 213.7 ± 10.2 | 372.4 ± 15.5 | 218.0 ± 1.0 | 329.2 ± 11.0 | 314.6 ± 3.4 | 319.8 ± 3.7 |
| IDQN | 193.8 ± 17.0 | 262.0 ± 10.1 | 161.7 ± 22.9 | 263.1 ± 4.8 | 230.5 ± 8.2 | 197.7 ± 12.0 |
| IDRQN | 199.1 ± 16.8 | 343.7 ± 52.4 | 189.7 ± 24.5 | 299.0 ± 7.5 | 276.9 ± 11.6 | 249.2 ± 13.5 |
| QMIX | 200.9 ± 18.1 | 313.4 ± 50.6 | 180.5 ± 28.7 | 262.3 ± 14.3 | 245.9 ± 25.9 | 189.2 ± 10.4 |

Table 7: Results at the end of training, on $200k$ and $2M$ time-steps for *Turb3Row1* and *Ablaincourt* respectively. *Sc. 1* (resp *Sc. 2*) corresponds to the firts Wind Scenario I (resp. II). All models are evaluated on both scenarios. For a decomposition in power and load penalty contribution, see Table 9 and Table 8.

| | Turb3Row1 | | Ablaincourt | |
| | FLORIS | FAST.Farm | FLORIS | FAST.Farm |
|---|---|---|---|---|
| | | Training on Sc. 1 | | |
| IPPO | 190.3 ± 2.0 | 239.6 ± 1.0 | 222.6 ± 5.4 | 245.0 ± 0.8 |
| MAPPO | 194.3 ± 1.9 | 237.5 ± 0.5 | 218.6 ± 5.8 | 239.8 ± 1.5 |
| IDQN | 159.1 ± 1.0 | 128.3 ± 0.7 | 157.6 ± 1.7 | 126.8 ± 1.7 |
| IDRQN | 160.5 ± 5.7 | 129.0 ± 4.7 | 168.1 ± 22.6 | 126.8 ± 1.2 |
| QMIX | 159.6 ± 2.2 | 128.7 ± 2.1 | 157.0 ± 0.5 | 126.3 ± 0.5 |
| | | Training on Sc. 2 | | |
| IPPO | **247.1** ± 7.1 | **251.6** ± 2.7 | **251.4** ± 0.5 | **253.4** ± 0.8 |
| MAPPO | 219.0 ± 32.7 | 248.8 ± 2.5 | 224.8 ± 20.5 | 247.6 ± 0.3 |
| IDQN | 159.6 ± 0.0 | 128.7 ± 0.0 | 159.4 ± 0.5 | 128.3 ± 0.7 |
| IDRQN | 176.1 ± 38.0 | 151.2 ± 51.6 | 176.2 ± 37.9 | 152.0 ± 51.2 |
| QMIX | 196.6 ± 45.2 | 179.3 ± 61.3 | 177.3 ± 36.5 | 127.6 ± 1.1 |

Table 8: Sum of total power output at every time-step for an evaluation episode with constant wind conditions (*Sc. 1*) at the end of training, after $200k$ time-steps for *Turb3Row1* and $2M$ time-steps for *Ablaincourt*.

| | Turb3Row1 | | Ablaincourt | |
|---|---|---|---|---|
| | **FLORIS** | **FAST.Farm** | **FLORIS** | **FAST.Farm** |
| | | Training on *Sc. 1* | | |
| IPPO | $84.0 \pm 0.9$ | $343.6 \pm 3.4$ | $82.3 \pm 1.0$ | $350.5 \pm 0.5$ |
| MAPPO | $84.7 \pm 0.7$ | $345.5 \pm 2.9$ | $82.9 \pm 0.8$ | $348.3 \pm 0.8$ |
| IDQN | $83.9 \pm 0.1$ | $\mathbf{285.7} \pm 0.4$ | $84.0 \pm 0.2$ | $284.9 \pm 0.9$ |
| IDRQN | $83.7 \pm 0.6$ | $286.1 \pm 2.5$ | $84.0 \pm 0.2$ | $284.9 \pm 0.6$ |
| QMIX | $83.8 \pm 0.2$ | $285.9 \pm 1.1$ | $84.1 \pm 0.0$ | $\mathbf{284.7} \pm 0.3$ |
| | | Training on *Sc. 2* | | |
| IPPO | $\mathbf{75.5} \pm 2.2$ | $351.7 \pm 2.0$ | $\mathbf{74.2} \pm 0.3$ | $352.5 \pm 0.8$ |
| MAPPO | $78.6 \pm 3.2$ | $349.5 \pm 2.3$ | $79.1 \pm 2.1$ | $350.6 \pm 2.1$ |
| IDRQN | $82.0 \pm 4.1$ | $297.9 \pm 27.5$ | $82.0 \pm 4.1$ | $298.3 \pm 27.3$ |
| QMIX | $79.8 \pm 4.9$ | $312.8 \pm 32.7$ | $82.0 \pm 3.7$ | $285.4 \pm 0.6$ |

Table 9: Sum of load penalties for an evaluation episode with constant wind conditions (*Sc. 1*) after $200k$ time-steps for *Turb3Row1* and $2M$ time-steps for *Ablaincourt*.

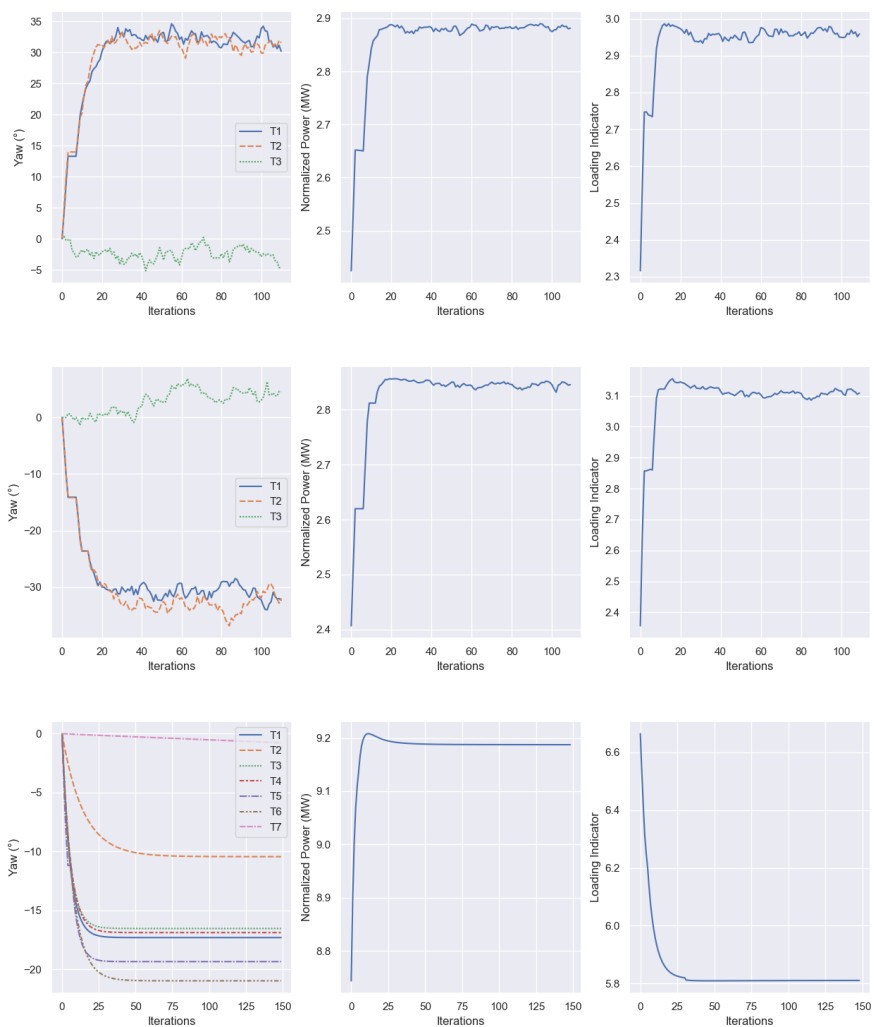

Figure 10: Behavior of 2 different yaw control policies learned on *Turb3Row1*, and a control policy learned on *Ablaincourt*, both on FLORIS environment with IPPO

# G    Wind farm as a graph

Knowledge of the farm layout and incoming wind direction can be exploited to represent wake interactions between wind turbines as a time-varying graph. In particular, under any given free-stream wind conditions, agent interaction structure can be modeled as a Directed Acyclic Graph. This is illustrated on Figure 11.

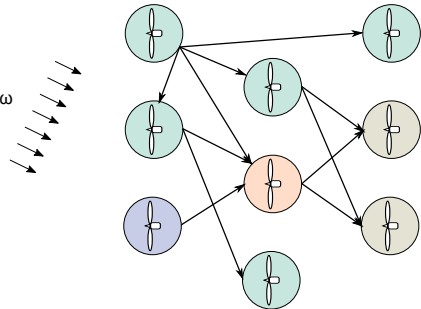

Figure 11: A wind turbine (purple) and its descendants in a wind turbine interaction DAG

# H    Visual Overview of Layouts

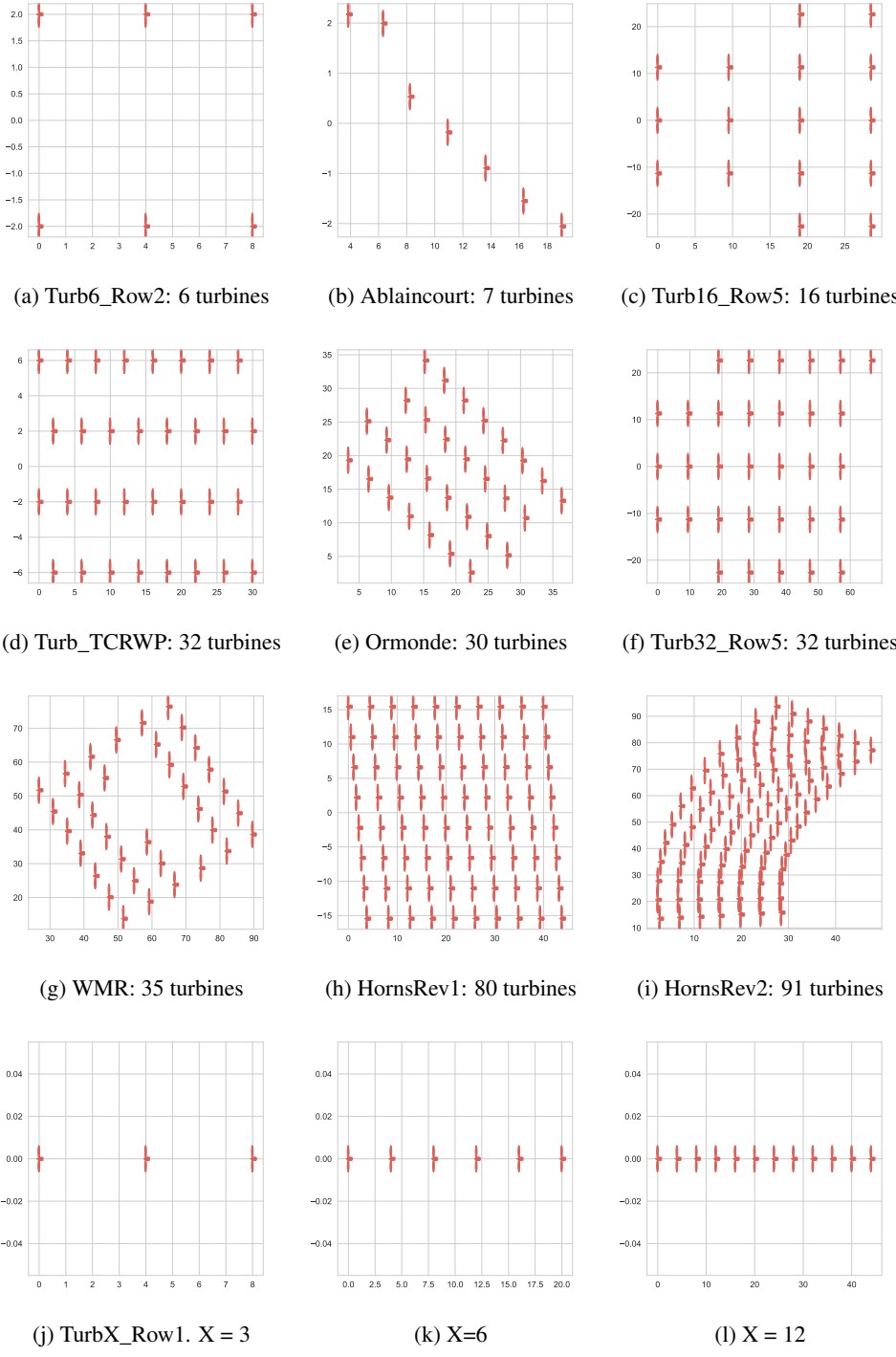

Figure 12: Coordinates of each wind turbine for the pre-registered layouts in WFCRL. Distances are in turbine diameters ($126m$ for the NREL 5MW Reference turbine). The *TurbX_Row1* toy layouts are procedurally generated for any value of $X$ between 1 and 12.

# I  Additional information on WFCRL

## I.1  List of dependencies

We report in the table below the list of open-source Python packages and other open-source software that WCFRL relies on.

| Software | License | License Link |
| --- | --- | --- |
| numpy | *Custom* | https://numpy.org/doc/stable/license.html |
| Gymnasium | MIT | https://github.com/Farama-Foundation/Gymnasium/blob/main/LICENSE |
| PettingZoo | MIT | https://github.com/Farama-Foundation/PettingZoo/blob/master/LICENSE |
| FLORIS | Apache v2.0 | https://github.com/NREL/floris/blob/main/LICENSE.txt |
| FAST.Farm (OpenFAST) | Apache v2.0 | https://github.com/OpenFAST/openfast/blob/main/LICENSE |
| mpi4py | *Custom* | https://github.com/erdc/mpi4py/blob/master/LICENSE.txt |
| Microsoft-MPI | MIT | https://github.com/microsoft/Microsoft-MPI/blob/master/LICENSE.txt |
| Open MPI | BSD 3-Clause | https://www.open-mpi.org/community/license.php |
| Seaborn | BSD 3-Clause | https://github.com/mwaskom/seaborn/blob/master/LICENSE.md |
| Matplotlib | *Custom* - BSD-compatible | https://matplotlib.org/stable/project/license.html |
| PyYAML | MIT | https://github.com/yaml/pyyaml/blob/main/LICENSE |
| Pandas | BSD 3-Clause | https://github.com/pandas-dev/pandas/blob/main/LICENSE |

## I.2  Licence

The WFCRL package is licensed under the Apache v2 license. The text of the license can be found here: https://github.com/ifpen/wfcrl-env/blob/main/LICENSE.

## I.3  Responsability

The authors bear all responsibility in case of violation of rights.

