# OpenReview forum: "WFCRL: A Multi-Agent Reinforcement Learning Benchmark for Wind Farm Control"
_NeurIPS.cc/2024/Datasets_and_Benchmarks_Track — NeurIPS 2024 Track Datasets and Benchmarks Poster_

### Official Review · Reviewer_i4kn · 2024-06-30
**Review to WFCRL: A Multi-Agent Reinforcement Learning Benchmark for Wind Farm Control**

**Rating:** 6
**Confidence:** 5
**Correctness:** Yes
**Clarity:** Yes

**Review:**

Strengths:
- The paper addresses a challenge in wind energy optimization, presenting a novel benchmark suite that can facilitate advancements in wind farm control using MARL.
- The paper provides clear and detailed descriptions of the simulators, wind farm layouts, and the MARL framework.

Weakness:
- Though "conventional control strategies suffer from the curse of dimensionality when the number of turbines increases", MARL also has this problem as it usually requires centralized training (the reward function in wind farm is also a function of global state and global action). It lacks discussion how to solve the scalability issue.
- By nature, wind farm suffers from wind speed uncertainties, but there is no discussion about how to solve the wind uncertainty for MARL.
- It is also an important concern for safety when the MARL strategies are deployed in a physical system. Won't it cause any damage to wind farms?
- There is not enough baseline in experiments. Please add more baselines like COMA, QMIX, QTRAN, RMA3C or other recent MARL algorithms.

**Strengths:**

- The paper addresses a challenge in wind energy optimization, presenting a novel benchmark suite that can facilitate advancements in wind farm control using MARL.
- The paper provides clear and detailed descriptions of the simulators, wind farm layouts, and the MARL framework.

**Additional Feedback:**

N/A

**Documentation:**

Yes

**Limitations:**

Yes

**Opportunities For Improvement:**

- Though "conventional control strategies suffer from the curse of dimensionality when the number of turbines increases", MARL also has this problem as it usually requires centralized training (the reward function in wind farm is also a function of global state and global action). It lacks discussion how to solve the scalability issue.
- By nature, wind farm suffers from wind speed uncertainties, but there is no discussion about how to solve the wind uncertainty for MARL.
- It is also an important concern for safety when the MARL strategies are deployed in a physical system. Won't it cause any damage to wind farms?
- There is not enough baseline in experiments. Please add more baselines like COMA, QMIX, QTRAN, RMA3C or other recent MARL algorithms.

**Relation To Prior Work:**

Yes

**Summary And Contributions:**

The paper introduces WFCRL (Wind Farm Control with Reinforcement Learning), a suite of multi-agent reinforcement learning (MARL) environments tailored for the wind farm control problem. WFCRL encompasses both static and dynamic simulators (FLORIS and FAST.Farm), providing a platform for testing and developing RL algorithms aimed at optimizing wind farm performance. The suite includes various wind farm layouts and offers interfaces for implementing transfer learning strategies from static to dynamic environments.

---

> ### Author Rebuttal · Authors · 2024-08-18
>
> Thank you for taking the time to review our paper, your feedback is very much appreciated. Please find in the following paragraphs our answers to your comments:
>
> > It lacks discussion how to solve the scalability issue.
>
> Indeed, although independent learning approaches allow to bypass the exponential increase of the global state, the reward function in our Dec-POMDP model is still a function of the global state/action.
>
> However, knowledge of the farm layout and incoming wind direction can be exploited to represent wake interactions between wind turbines as a time-varying graph. This approach allows local reward functions for decentralized learning RL algorithms, where agents communicate with their neighbors. We have recently presented current work on such an approach [4 (see Section 4 on application to wind farm control], that will be implemented in WFCRL.
>
> Note that WFCRL contains a `RewardShaper` class that allows to easily design custom reward functions, and can thus be used to train decentralized learning approaches.
>
> > There is no discussion about how to solve the wind uncertainty for MARL.
>
> We assume that local measurements of wind speed and direction are available at each wind turbine, as wind turbines are usually equipped with anemometers. On the FAST.Farm simulator, we estimate free-stream wind speed/direction with  the measurements of the wind turbine with the highest wind speed. Since inflow wind can be turbulent, all estimates are averaged over a period of time defined by the *buffer_window* parameter.
>
> We agree that this does not represent the complexity of wind speed estimation on real wind farms, in which turbine measurement devices are often located at a single point far below the rotor plane. One option is to simulate a wind speed estimation strategy in our environments, as recent works have developed accurate methods to estimate hub-height wind speed [1,2] from these measurements. This is hard to do in FAST.Farm, for which collection of wind estimation points requires an expensive file loading step at every time-step (see Issue [3]).
>
> We thus rather chose to position ourselves after the wind speed and direction estimation step, and assume our observations are accurate. The addition of random noise in observations can however be an interesting improvement to better reflect measurement errors.
>
> >  It is also an important concern for safety when the MARL strategies are deployed in a physical system. Won't it cause any damage to wind farms?
>
> Indeed, and this is why WFCRL allows to train wind control algorithms for load mitigation. Our proposed benchmark (**l. 235**) adds both a load penalty (**l.259, 264**) to the shared reward, and a hard constraint to maintain yaw actuation time below 10%. (**l. 245**).  Learned policies therefore face an explicit tradeoff between maximizing power and damaging wind turbines.
>
> > There is not enough baseline in experiments. Please add more baselines like COMA, QMIX, QTRAN, RMA3C or other recent MARL algorithms.
>
> We concur that more baselines should be added to WFCLR. Following your comments, we have since added a QMIX baseline to our benchmark repository (https://github.com/ifpen/wfcrl-benchmark), and will update the paper with the corresponding experiments.
>
> Thank you for your time. We hope that we have answered some of your concerns, and are happy to further discuss any of these points.
>
> [1] Chen P. _et al_, Effective wind speed estimation study of the wind turbine based on deep learning*, Energy, Volume 247, 2022 (https://www.sciencedirect.com/science/article/pii/S0360544222003942)
>
> [2] Liu, B _et al_,Estimating hub-height wind speed based on a (machine learning algorithm: implications for wind energy assessment, Atmos. Chem. Phys.,2023 (https://doi.org/10.5194/acp-23-3181-2023)
>
> [3] Github Issue: https://github.com/OpenFAST/openfast/discussions/2316
>
> [4]  Monroc, Claire Bizon _et al_, Wind farm control with cooperative multi-agent reinforcement learning, ICML 2024 Workshop: ARLET

---

### Official Review · Reviewer_Sh7C · 2024-07-09
**A scalable interfacing simulator for wind farm energy control**

**Rating:** 6
**Confidence:** 4

**Review:**

The work addresses a significant gap in the field by providing a standardized, open-source platform for researchers to develop and benchmark reinforcement learning algorithms for wind farm optimization. The authors provide a thorough explanation of the wind farm control problem, the challenges posed by wake effects, and the potential benefits of intelligent control strategies. The description of the MARL framework and its application to wind farm control is clear and well-reasoned. The inclusion of both power production and turbine fatigue loads in the reward function reflects real-world considerations in wind farm operation, adding to the practical relevance of the work.

Overall, in my opinion, the formulation is well-suited to the inherent structure of wind farms and allows for scalable, distributed control strategies. The multi fidelity simulator approach enables researchers to explore the trade-offs between computational efficiency and model fidelity, as well as investigate transfer learning strategies between simulators of different complexities. The inclusion of multiple layouts with realistic settings in some of them adds significant value and realism to the benchmark suite. One of the most innovative aspects of WFCRL is its potential for transfer learning between the static FLORIS and dynamic FAST.Farm simulators. This feature addresses a critical challenge in the field: bridging the gap between simplified models used for algorithm development and the complex dynamics of real-world wind farms. While the paper mentions this potential, a more in-depth exploration of transfer learning strategies and results would have further strengthened the contribution.

The computational complexity of dynamic simulations remains a significant challenge, as acknowledged by the authors. This limitation may impact the scalability of the proposed approach for larger wind farms or longer time horizons. Further research into methods for accelerating dynamic simulations or developing intermediate-fidelity models could be valuable extensions of this work. Another limitation is the lack of extensive validation against real-world data. For figures and tables highlighting the results, some improvements in labeling, consistency of notation (eg state space description table) and explanations should be done.

I briefly looked at the codebase and the environment suite is sufficiently customizable, allowing researchers to design and run their own scenarios for both centralized and multi-agent RL approaches. The documentation of the environment structure(from wfcrl-env and wfcrl-env-benchmark), observation and action spaces, and reward formulations will facilitate adoption by other researchers in the field.

**Strengths:**

WFCRL provides a wide range of wind farm layouts and wind condition scenarios, allowing researchers to test algorithms under various realistic conditions. The inclusion of both static (FLORIS) and dynamic (FAST.Farm) simulators is particularly valuable for evaluating algorithm performance across different levels of model fidelity.The setup of the multiprocessing interface for FAST.Farm (from fig 5) is a good work done by the authors.
The framing of wind farm control as a cooperative MARL problem is well-suited to the inherent structure of wind farms. This approach allows for scalable and distributed control strategies. The benchmark example incorporates both power production and turbine fatigue loads in the reward function. The inclusion of both FLORIS and FAST.Farm simulators enables research into transfer learning strategies. The authors provide a good overview of related work in wind farm control and reinforcement learning applications. They clearly position WFCRL as addressing a gap in the field. The paper builds upon established wind farm simulators (from NREL) and MARL algorithms(from Eugene Vinitsky's work), extending their application to create a comprehensive benchmark suite.

**Additional Feedback:**

None

**Clarity:**

Overall, the paper is well-structured and clearly written. The technical details are thoroughly explained, making it accessible to researchers familiar with reinforcement learning and wind energy. However, some sections, particularly those describing the simulators and state space description, notations, could benefit from more concise explanations or additional visual aids.

**Correctness:**

The paper appears to be technically correct in its formulations and implementations. The Dec-POMDP framework is appropriately applied to the wind farm control problem, and the implementations of IPPO and MAPPO seem to follow standard practices. The use of established simulators (FLORIS and FAST.Farm) adds credibility to the results.

**Documentation:**

There is sufficient documentation on the code side

**Limitations:**

Computational complexity: The authors acknowledge that "dynamic wind farm simulators remain slow" (line 298). This limitation may impact the scalability of the proposed approach for larger wind farms or longer time horizons. While FAST.Farm provides higher fidelity simulations, there is an ongoing challenge of balancing model accuracy with computational efficiency. The paper primarily focuses on simulated environments. While based on real wind farm layouts, there is limited discussion of how well the simulated results align with real-world wind farm performance.

**Opportunities For Improvement:**

While the paper implements IPPO and MAPPO, it could benefit from a broader comparison with other MARL algorithms, such as QMIX or MADDPG, to provide a more comprehensive benchmark. Especially since the authors mention that they have agents with different observation The paper mentions the potential for transfer learning between FLORIS and FAST.Farm, but does not provide extensive results or analysis in this area. Expanding on this aspect could strengthen the paper's contributions.The wind condition scenarios could be further refined to include more complex temporal patterns or spatial variations across the wind farm.

A few individual points:
*In line 231, you say you adapt CleanRL, why do you not directly adapt the code with the work in [39]
*How do they authors balance the power generation vs fatigue prevention balance in the reward. How is the default value 0.1 chosen?
* In the paragraph at L171 and table 2, please fix the inconsistencies in state space description as well as the notation
*L232, authors discuss about local observations are not identical, hence they do not implement weight sharing. In that regard have the authors looked at heterogeneous multiagent RL implementations?
*L270: Are the weights for rho exhaustive ie it covers all possible wind conditions?
*It is not clear from the diagram 7 appendix F.1 whether transfer learning or only evaluation leads to better results than greedy baseline. In this regard the description of figure 7 should be improved.

**Relation To Prior Work:**

Please see strengths section where I briefly discuss how the authors position the work

**Summary And Contributions:**

This paper introduces WFCRL (Wind Farm Control with Reinforcement Learning), a suite of multi-agent reinforcement learning environments for wind farm control. The work interfaces with two wind farm simulators: the static FLORIS and dynamic FAST.Farm, allowing for transfer learning between simulators of different fidelities. It implements 10 wind farm layouts, including 5 real wind farms, for each simulator. The paper frames the wind farm control problem as a Decentralized Partially Observable Markov Decision Process (Dec-POMDP) with M interacting agents. They provide detailed specifications of the state space, observation spaces, action spaces, and reward functions for both FLORIS and FAST.Farm environments. The benchmark example demonstrates the application of IPPO and MAPPO algorithms to maximize total power production while considering turbine fatigue loads.

---

> ### Author Rebuttal · Authors · 2024-08-18
>
> We thank you for taking the time to review our paper. Your thorough feedback is very much appreciated, and you can find below our answers to your comments:
>
> > Wind condition scenarios
>
> We have added support for wind time-series. Any time series can be provided by the user, and we propose to use real measurements from SMARTEOLE dataset  (see  https://github.com/ifpen/wfcrl-env/).
>
> >  Comparison with other MARL algorithms
>
> Following your comments, we have added a QMIX baseline to our benchmark repository (https://github.com/ifpen/wfcrl-benchmark), and will update the paper with the corresponding experiments.
>
> >  Lack of extensive validation against real-world data.
>
> FAST.Farm has been validated against both real wind farm data [1, 3] and Large Eddy Simulations [2,4].  Power predictions match real measurements within 2-7% error on average [1], although the error can reach 18% for downstream turbines under low free-stream wind speeds. Load predictions are within 10% of groundtruth values, with errors reaching 25% for certain wind directions [4]. Predictions always respect *trends* in power/load changes: under the same wind conditions, yaws that lead to increases in power output in simulations also lead to increases in real wind farms.
>
> To the best of our knowledge, an explicit investigation of the transfer of yaw optimization results from FAST.Farm simulations to a real wind farm has not yet been carried out, and we therefore cannot quantify this discrepancy. We can modify the paper to mention it under Limitations.
>
> > Accelerating dynamic simulations or developing intermediate-fidelity models.
>
> Indeed, WFCRL would greatly benefit from faster dynamic models, and its interfacing approach makes it easy to pair it with any wind farm simulator.  Although this development is currently out of scope of this work, it can be considered for future improvements.
>
> > Why do you not directly adapt the code with the work in [39]
>
> The choice to adapt CleanRL codes was made to build on its simple and single-page implementations. To the greatest extent possible we have tried to maintain this feature: this makes our code easy to modify by others, and is in line with our goal of building a bridge between the RL and wind energy community.
>
> > How is the default value 0.1 chosen?
>
> The default value of 0.1 is chosen in order to ensure the load component of the reward is of the same magnitude as the production component, and both are given similar weight with respect to return maximization. This value can however be changed to give more importance to one or the other.
>
> > Heterogeneous MARL
>
> Heterogenous multi-agent RL implementations are indeed relevant for our problem, and we also think that wind turbines' unique identification by their coordinates in the farm could also be exploited to learn efficient representations of each agent. We have for now decided to restrict the scope of this paper to simpler alternatives, but this is an interesting path for future baselines.
>
> > Are the weights for rho exhaustive ?
>
> The weights for rho cover all possible wind directions (0 to 360°), as well as all wind speeds between cuti-in and cut-off speed of the wind turbine. For wind speeds above and below, the wind turbine shuts down to prevent damage.
>
>  >  Description of figure 7, and more in-depth exploration of transfer learning
>
> The diagram 7 in appendix F.1 represents the percentage increase over the greedy baseline, of both power output and load under 2 strategies. The average is taken over 5 experiments launched on 5 seeds, and the standard deviation is reported on the graph. Under the first strategy (blue line on the Figure), policies learned on FLORIS are deployed on the dynamic simulator, and achieve an increase of 15% in power production over the baseline. On Turb3_Row1, we know that there exists an optimal policy that reaches an increase of 30%. The difference in performance suggests that we could benefit from further fine-tuning these policies on the dynamic simulation. However, the transfer learning strategy (orange line on the Figure)  degrades the performance of the policies when learning online, reaching an average of 0% increase over the baseline at the end of the experiments.
>
> Our goal here is to illustrate the challenge of designing robust methods to bridge the gap between simple simulators and complex real world dynamics. We acknowledge that the description of this figure should have been more detailed, and commit ourselves to update it.
>
> Moreover, a more thorough investigation in transfer strategies and adaptation from simple to complex models can be carried out. In particular, we have in a previous work investigated fine-tuning on FAST.Farm simulations policies learned by imitation with static models [5], and can update the paper with an evaluation of this approach on WFCRL.
>
> We also thank you for your comments on the readability of figure legends, as well as the error in notation **L171**/** table 2**, we will update our paper accordingly.
>
> We hope that we have been able to address some of your concerns, and are available to further discuss any of these points.
>
> [1] Shaler K.  _et al_  , Validation of FAST.Farm Against Full-Scale Turbine SCADA Data for a Small Wind Farm, _J. Phys.: Conf. Series, 2020
>
> [2] Jonkman,  J _et al_ , Validation of FAST.Farm Against Large-Eddy Simulations, J  _J. Phys.: Conf. Series, 2018
>
> [3] Kretschmer, M. _et al_, FAST.Farm load validation for single wake situations at alpha ventus, Wind Energ. Sci., 2021
>
> [4] Shaler K. _et al_ , FAST.Farm development and validation of structural load prediction against large eddy simulations, 2020
>
> [5] Bizon Monroc C _et al_, Towards fine tuning wake steering policies in the field: an imitation-based approach, 2024

---

> > ### Comment · Reviewer_Sh7C · 2024-08-26
> > **I am maintaining my current rating.**
> >
> > I have gone through your rebuttal. I am maintaining my current rating.

---

### Official Review · Reviewer_MFL8 · 2024-07-11
**Review of paper 1105**

**Rating:** 8
**Confidence:** 5
**Correctness:** The claims are correct.

**Review:**

The paper is well written, easy to understand, and the results are clearly explained. Contributions and limitations are also well described.

**Strengths:**

Besides the clarity of the paper, the strength lies in the development of the simulator for a topic of high interest. Moreover, various variants of the components of the benchmarking tool are possible according to one's needs.

**Additional Feedback:**

NA

**Clarity:**

The paper is really well written. There is however a typo in the title, as well as some typos in the text.

**Documentation:**

Yes.

**Ethics:**

No issue.

**Limitations:**

The limitations are clearly explained, but rather short. Aren't there any other limitations besides the speed of the simulators?

**Opportunities For Improvement:**

some suggestions:
- Is there a link with Infrastructure Management Planning tools & framework?
- In the list of contributions, I would mention the suite of environments is open (as far as I got it).
- In lines 30 to 41, I guess some figures would help understand the claims.

**Relation To Prior Work:**

Relation to prior work is clearly explained.

**Summary And Contributions:**

The authors present a benchmark for wind farm control suing multi-agent RL.In particular, two siulators are possible to use and have been tested, as well as various MARL algorithms.

---

> ### Author Rebuttal · Authors · 2024-08-18
>
> Thank you very much for taking the time to review our paper and for your positive feedback. We hereafter respond to your comments:
>
> >   Is there a link with Infrastructure Management Planning tools & framework?
>
> The case of wind farms has indeed been considered in the Management Planning tools & framework [1], with the goal of optimizing inspections and repairs over the farm's lifetime.  The focus there is therefore on modeling the probability of failure for each component.
>
> Following this approach, WFCRL could use our load estimates to directly model failure events , but this would require modeling the wind farm at a very different time scale (the life time of a turbine) than the one currently considered. Similarly, load estimation outputs from WFCRL could be fed as inputs to an IMP optimization problem, which would consider the choice of the control algorithm as an action.
>
> These interesting connections can be explored in the future,  although we think they are out of the scope of this paper.
>
> > In the list of contributions, I would mention the suite of environments is open (as far as I got it).
>
> Thank you for this comment. WFCRL is indeed complety open-source, and we will update our list of contributions to make this fact clearer.
>
> >    In lines 30 to 41, I guess some figures would help understand the claims.
>
> Thank you very much for this feedback. Could we ask you to be clearer about which of the claims made in this paragraph you think should be illustrated ?
>
> For the claim about the graph structure of the problem with regard to the wake interactions between turbines, we provide a visual explanation in the additional PDF. We would gladly provide additional figures with a more detailed schema of dynamic interactions between turbines and/or illustrate other claims if necessary.
>
> > The limitations are clearly explained, but rather short. Aren't there any other limitations besides the speed of the simulators?
>
> We think the speed of mid- or high- fidelity simulators is the main constraint when designing and evaluating scalable strategies that can be deployed on real wind farms.
>
> Another important limitation is that our proposition to evaluate transfer methods relies on the assumption that ability to transfer from static to dynamic models is a good proxy for robustness in the real world. We believe this assumption to be reasonable as FAST.Farm has been extensively validated against real data. However, the proposition that selecting for transfer capacity produces better policies in the real world has so far never been tested for wind farm control.
>
> [1] Leroy P. _et al_, IMP-MARL: a Suite of Environments for Large-scale Infrastructure Management Planning via MARL, Advances in Neural Information Processing Systems, 2024

---

### Decision · Program_Chairs · 2024-09-26

**Decision:**

Accept (Poster)

**Comment:**

This paper creates a novel suite of multi-agent reinforcement learning environments for the wind farm control problem. This is an interesting and significant contribution to the research field of multi-agent reinforcement learning and the application area of wind farm control. The paper is well written.
All reviewers think that this paper is acceptable, and one reviewer strongly recommends the acceptance.
I think that this paper is acceptable.
As pointed out by one reviewer, there exists a tradeoff relation between model accuracy and computational efficiency. As a result, it is likely that the use of intermediate fidelity models is needed for some real-world application tasks whereas high fidelity models can be used for other real-world application tasks. It would increase the usefulness of the proposed benchmarking suite if the authors could give some additional comments/discussions about an appropriate tradeoff for the real-world wind farm layouts used in this paper.